

# Analysis of Wave Propagation in a Discrete Chain of Bilinear Oscillators

Maria S. Kuznetsova[1], Elena Pasternak[2], Arcady V. Dyskin[1]

[1]School of Civil, Environmental and Mining Engineering, The University of Western Australia, Perth, 6009, Australia
[2]School of Mechanical and Chemical Engineering, The University of Western Australia, Perth, 6009, Australia

*Correspondence to*: Maria S. Kuznetsova (maria.kuznetsova@research.uwa.edu.au)

**Abstract.** The process of wave propagation in the discrete chain of bilinear oscillators subjected to several types of external harmonic excitation has been analysed. The phenomenon of sign inversion of the displacement is observed for tension-compression excitation. The solution for wave propagation in a continuous 1D bimodular rod is developed and the numerical results are compared.

## 1 Introduction

In this paper, we analyse the process of wave propagation in a chain of bilinear oscillators – discrete masses connected by springs having different stiffnesses in tension and compression. Due to their simplicity, discrete chains of bilinear oscillators have often been used in the problems related to non-linear vibrations of mechanical systems, such as vibrations in suspension bridges (De Freitas and Viana, 2004) and in the systems with the so-called fatigue cracks (Rivola and White, 1998; Ohara et al., 2007; Peng et al., 2008). Bilinear oscillators were also used in mathematical modelling of seismic isolation systems (Skinner et al. 1993; Chang et al., 2002).

The behaviour of the bilinear oscillators has been recently studied in (Dyskin et al., 2012; Dyskin et al., 2014; Guzek et al., 2016) for a limiting case of an infinite stiffness in compression. However, a general case of a discrete chain of bilinear oscillators has never been studied with respect to the mechanical wave propagation, which is why it has been decided to numerically investigate the response of the bilinear system that could represent a continuous bimodular media.

We focus on a conservative system; for the effects of damping in bilinear oscillators see (Shaw and Holmes, 1983; Natsiavas, 1990a, 1990b; Liu et al., 2015; Dyskin et al., 2012; Klepka et al., 2015, Guzek et al., 2016).

The purpose of the present work is to study the response of a discrete system of bilinear oscillators loaded by an external harmonic force. We also developed a solution for wave propagation in a continuous 1D bimodular rod for further comparison.





## 2 Mathematical formulation

We consider an infinite chain of masses and bilinear springs, where masses $m$ are supposed to be identical, springs have the

length $l$ and the stiffness described in the following formula

$$K(U) = \begin{cases} K_t & \text{for } \Delta U \geq 0 \\ K_c & \text{for } \Delta U < 0 \end{cases}, \tag{1}$$

Here, $U(X,T)$ is the displacement and $\Delta U$ is the difference of displacement of two adjacent masses, that is the displacement

of each spring. The mass-spring chain is fixed at the right end (Fig.1) and loaded by an external force $F(t)$ from the left end.

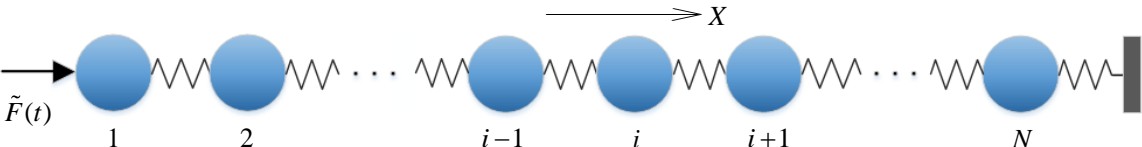

**Figure 1: Elastic mass-spring chain.**

By introducing the Lagrangian $L = T - V$, which is the difference between the total kinetic and strain energy of the system

where

$$T = \Sigma \frac{1}{2} M \dot{U}_i \text{ and } V = \Sigma \frac{1}{2} K_i \left( U_{i+1} - U_i \right)^2, \; i = 1..N, \tag{2}$$

We obtain the governing equation of the longitudinal motion of $i$-th mass

$$M \ddot{U}_i + \left( K_i + K_{i-1} \right) U_i - K_{i-1} U_{i-1} - K_i U_{i+1} = \tilde{F}_i \tag{3}$$

where $i = 1$ corresponds to the first mass from the left end. Since loading is applied to the left end, it follows that

$$\tilde{F}_i = \begin{cases} \tilde{F}(t) & \text{for } i = 1 \\ 0 & \text{otherwise} \end{cases}. \tag{4}$$

We rewrite the equation of motion (3) in terms of dimensionless displacement $u$ and dimensionless time $t$

$$u = \frac{\Omega}{c} U, \quad t = \Omega T, \tag{5}$$

where $\Omega$ is the characteristic frequency associated with the duration of the applied external force $\Omega = \frac{2\pi}{T_0}$, $c$ is the sound

velocity in the discrete chain $c = l \sqrt{\frac{K}{M}}$. This yields

$$\ddot{u}_i + \left( k_i + k_{i-1} \right) u_i - k_{i-1} u_{i-1} - k_i u_{i+1} = F_i \tag{6}$$





Here, $F_i$ and $k_i$ are the dimensionless forces and stiffnesses of the springs, respectively:

$$F_i = \frac{\tilde{F}_i}{M\Omega c} \,, k_i = \frac{K_i}{M\Omega^2} \tag{7}$$

Without loss of generality, we adopt that the springs are stiffer in compression, and by introducing the stiffness ratio $a$, obtain higher compressive stiffness $k_c = 1 + a$ and lower tensile stiffness $k_t = 1 - a$.

The system is initially assumed to be at rest, i.e. $u_i(0) = \dot{u}_i(0) = 0$.

## 3 Mechanical parameters of the discrete mass-spring chain

All the numerical results presented in the paper are obtained for the following dimensional and dimensionless parameters:

|  | Notation | Value |
| --- | --- | --- |
| Total number of masses | $N$ | 100 |
| Length of the spring | $l$ | 1 m |
| Stiffness ratio | $a$ | $\frac{1}{3}$ |
| Amplitude of the applied force | $F_0$ | $10^{-4}$ |
| Frequency of the applied force | $\omega$ | 0.25 |

## 4 Impulse harmonic excitation

In the analysis of wave propagation caused by initial excitation, simple harmonic or sinusoidal waves are of substantial interest. Due to its simplicity, let us analyse the case of a harmonic impulse first. The external loading subjected to the left end of the chain and is described as follows

$$F(t) = \pm F_0 H\left(\frac{2\pi}{\omega} - t\right)\sin(\omega t) \tag{8}$$

### 4.1 Compression-tension harmonic impulse

The analysis will start with the positive sign in Eq. (8), in other words when compression is followed by tension. Knowing that in the bimodular chain the compressive wave travels with a higher speed than the tensile one, one would expect the distance between compressive and tensile zones increasing with time. Figure 2 depicts the displacement field along the bilinear chain against the mass number (integer value of the coordinate) at different time moments. Since the initial load is applied from the left end of the chain, the positive displacement corresponds to compression and negative one to tension. As expected, looking at the zones with zero deformation, i.e. horizontal regions with nearly constant positive displacement, makes it clear that the





gap between compressive and tensile fronts elongates with time. This phenomenon always takes place when the external excitation corresponding to a faster wave is followed by a slower one.

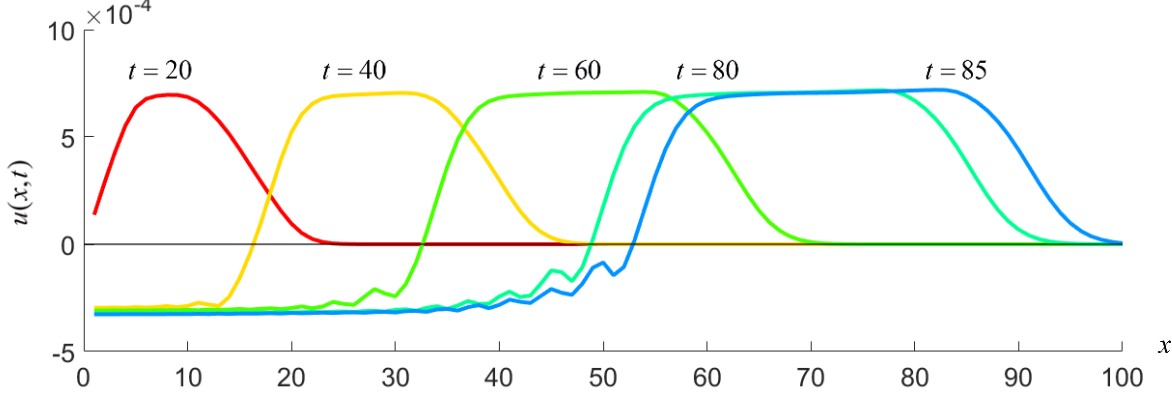

**Figure 2: Displacement** $u(x,t)$ **at different time moments versus the horizontal coordinate** $x$ **for the compression-tension harmonic**
**impulse.**

### 4.2 Tension-compression harmonic impulse

The second type of loading is described by Eq. (8) taken with a negative sign. This case is of considerable interest due to the fact that excitation corresponding to a slower wave speed is followed by a faster one. In this case, the faster wave front catches up with a slower front, which leads to an unusual behaviour of displacement observed in Fig. 3. Soon after the collision $t*$
between the compressive and tensile wave fronts, the displacement gradually changes from negative to positive implying that, although a tensile impulse is applied first, the system undergoes compressive displacement after the collision. Hereafter this phenomenon is called the sign inversion.

The collision is defined by the time when the fast moving wave front with negative gradient touches the wave front with the slow moving positive gradient and is determined from the following equation

$$c_t\left(t^* - \frac{\pi}{\omega}\right) = c_c\left(t^* - \frac{2\pi}{\omega}\right) \tag{9}$$

which gives $t^* \approx 55$ for this particular case.

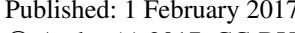
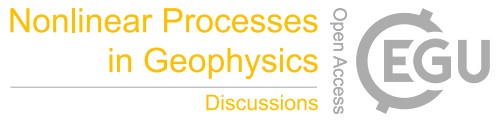


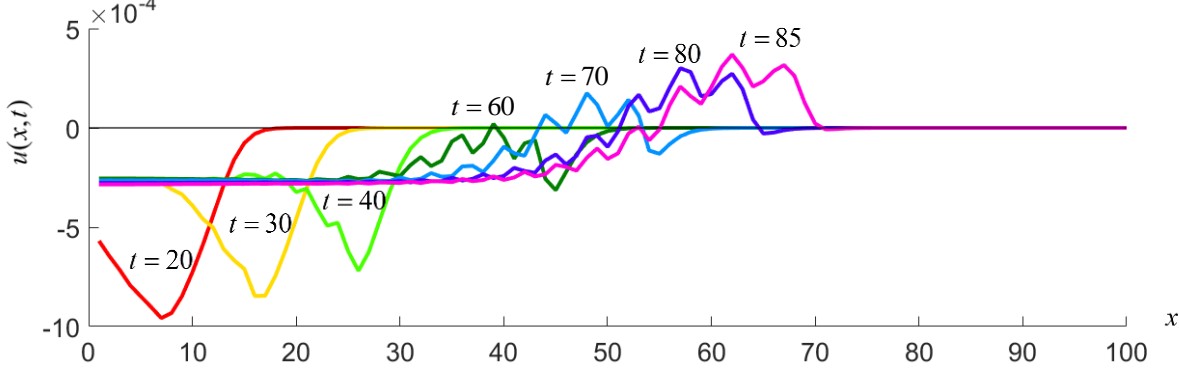

**Figure 3: Displacement** $u(x,t)$ **at different time moments versus the horizontal coordinate** $x$ **for the tension-compression harmonic impulse.**

### 4.3 Energy conservation

As an additional check on the accuracy of the numerical solution, the integral total energy $E = T + V$ has been calculated for the entire system of masses and bilinear springs. As seen in Fig. 4, soon after the impulse loading is applied (that is energy is added to the system), the total energy reaches its maximum and remains constant throughout the entire solution.

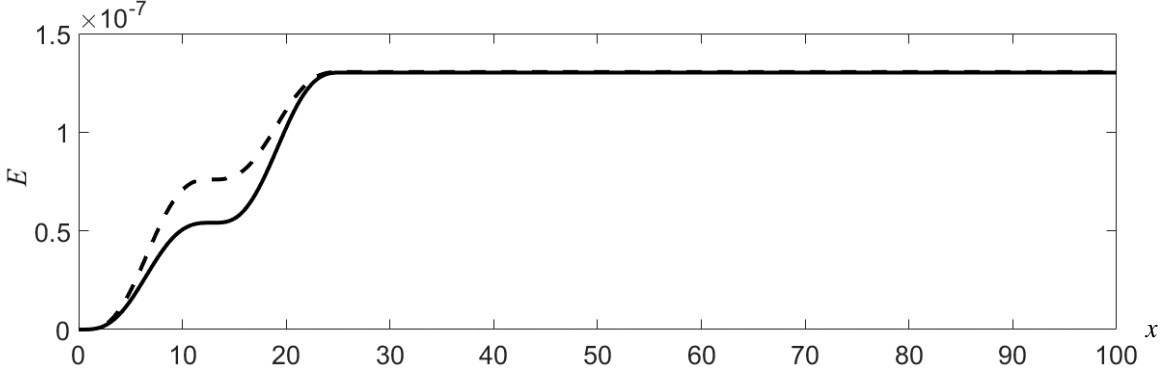

**Figure 4: Intergral total energy in the discrete chain with respect to time: for compression-tension (the solid line) and tension-**
**compression (the dashed line) harmonic impulses.**

### 5 Continuous harmonic excitation

The second type of excitation considered here is a continuous external loading applied to the left end of the chain:

$$F(t) = \pm F_0 \sin(\omega t) \tag{10}$$

As in Sect. 4, two cases will be considered: compression-tension and tension-compression sequences. Obviously, in the case

of continuous excitation, the difference between these two cases is the difference in the initial phase.



## 5.1 Compression-tension harmonic excitation

Numerical solution for displacement $u$ at different times versus horizontal coordinate $x$ is presented in Fig. 5. It may be observed that, due to the tensile stiffness being lower than the compressive one, the displacements close to the left end of the chain decrease with time, implying that the left part of the chain undergoes increasing tensile displacements.

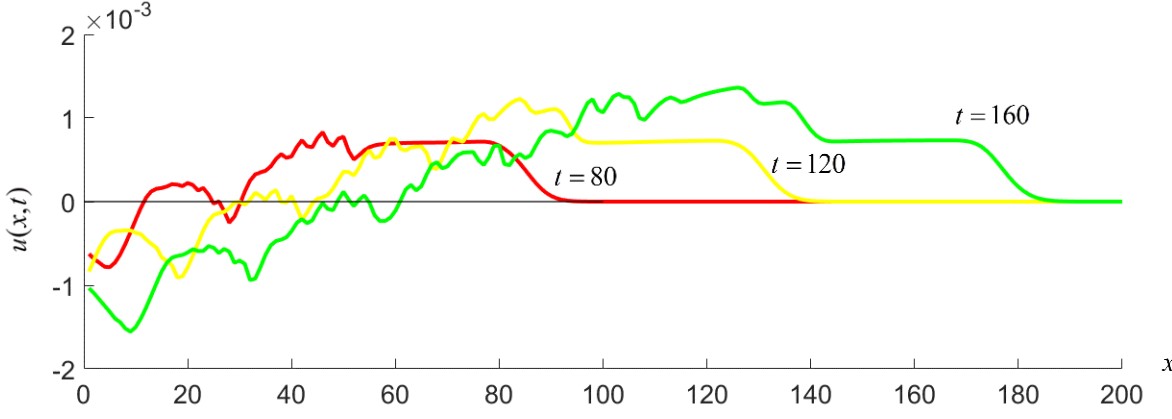

**Figure 5: Displacement $u(x,t)$ at different time moments versus the horizontal coordinate $x$ for the compression-tension harmonic excitation.**

## 5.2 Tension-compression harmonic excitation

Figure 6 represents displacement $u$ along the discrete chain at different time moments. Comparison of Figs. 5, 6 suggests that the numerical solution exhibits little sensitivity towards the excitation phase. This is easy to interpret given that the compression-tension and tension-compression excitations are just different phase shifts of the same continuous harmonic excitation.

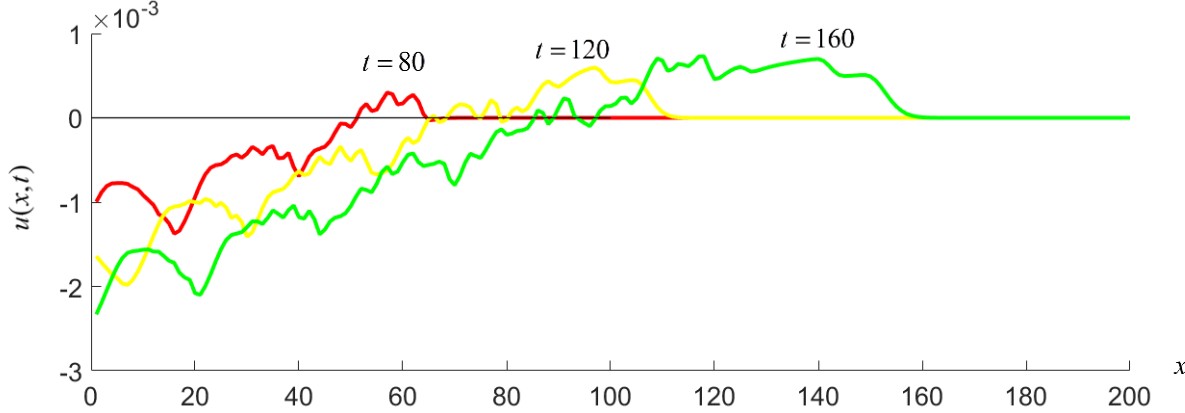

**Figure 6: Displacement $u(x,t)$ at different time moments versus the horizontal coordinate $x$ for the tension-compression harmonic excitation.**



## 6 Comparison with another numerical model and analytical solution

In this section, we want to compare the numerical results for the discrete chain of bilinear oscillators with those for a 1D bimodular rod subjected to the same boundary conditions. The wave equation for a 1D rod made of a bimodular material reads

$$\left(E - e\operatorname{sgn}\left(\frac{\partial U}{\partial X}\right)\right)\frac{\partial^2 U}{\partial X^2} = \rho\frac{\partial^2 U}{\partial T^2} \tag{11}$$

where $U, X, T$ are displacement, coordinate along the rod and time respectively, $\rho$ is the specific mass, $E$ is "average" Young's modulus, and $e$ is the difference between Young's moduli in tension $E_t = E - e$ and in compression $E_c = E + e$.

In the dimensionless form, Eq. (11) reads

$$\left(1 - a\operatorname{sgn}\left(\frac{\partial u}{\partial x}\right)\right)\frac{\partial^2 u}{\partial x^2} = \frac{\partial^2 u}{\partial t^2} \tag{12}$$

where $x = \dfrac{\Omega}{\sqrt{E/\rho}}X;\quad t = \Omega T;\quad a = \dfrac{e}{E};\quad u = \dfrac{\Omega}{\sqrt{E/\rho}}U;\quad \Omega = \dfrac{2\pi}{T_0}$ and $T_0$ is the duration of the applied external impulse.

The analytical solution for the compression-tension excitation of the frequency $\omega = 1$ described in Sect. 6.1 has been derived in (Gavrilov and Herman, 2012) and was later extended for the arbitrary excitation frequency in (Kuznetsova et al., 2016). Numerical results are obtained by solving Eq. (12) numerically using the central difference scheme.

To match the results obtained for the discrete chain of bilinear oscillators, spatial and time discretisation is chosen to be the same ($\Delta x = 1$ and $\Delta t = 10^{-3}$, respectively) and all other parameters being used from the Table 1.

### 6.1 Compression-tension harmonic impulse

The displacements $u(x,t)$ for various times are plotted in Fig. 7, which includes the analytical solution (bold dashed line) and numerical results for the discrete chain and the 1D bimodular rod (solid and dash-dot lines, respectively). One can observe that the three approaches show good agreement at the wave front and a slight discrepancy behind the wave front, which is typical for the second-order finite difference schemes (Kukudzhanov, 2013).



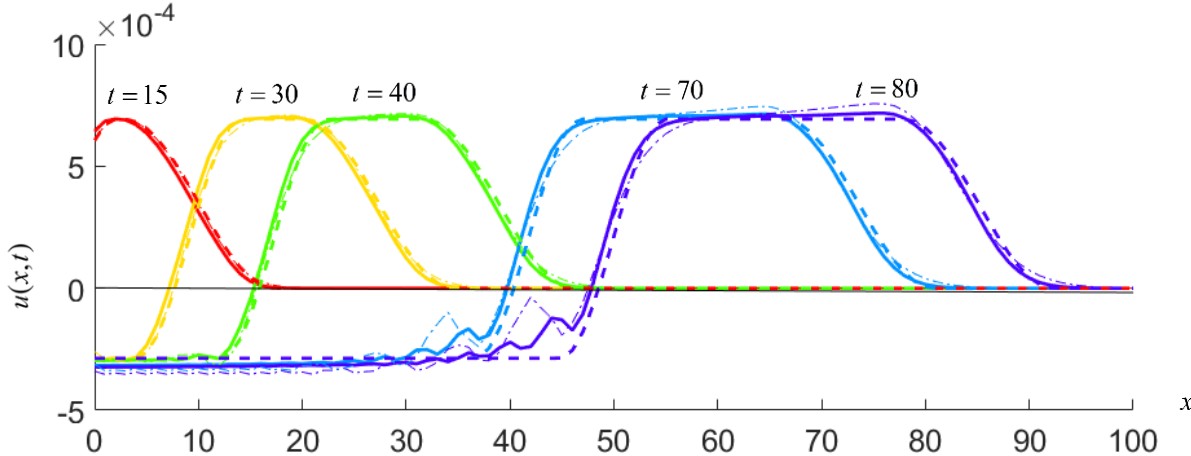

**Figure 7: Displacement $u(x,t)$ at different time moments versus the horizontal coordinate $x$ for the compression-tension harmonic impulse: analytical solution (dashed line), numerical solutions for the discrete chain (solid line) and the bimodular rod (thin dash-dot line).**

## 6.2 Tension-compression harmonic impulse

As the analytical solution does not exist for this case, only numerical results are presented. Figure 8 shows the numerical results for the displacements $u(x,t)$ for the discrete chain (solid line) and the 1D bimodular rod (dash-dot line). It is interesting to note that with all parameters being equal, the discrete chain generally exhibits lower displacements throughout the entire solution. This discrepancy may be explained by the insufficiently small spatial step for the rod since it is assumed it to be equal to the length of the springs in the discrete chain which equals 1.

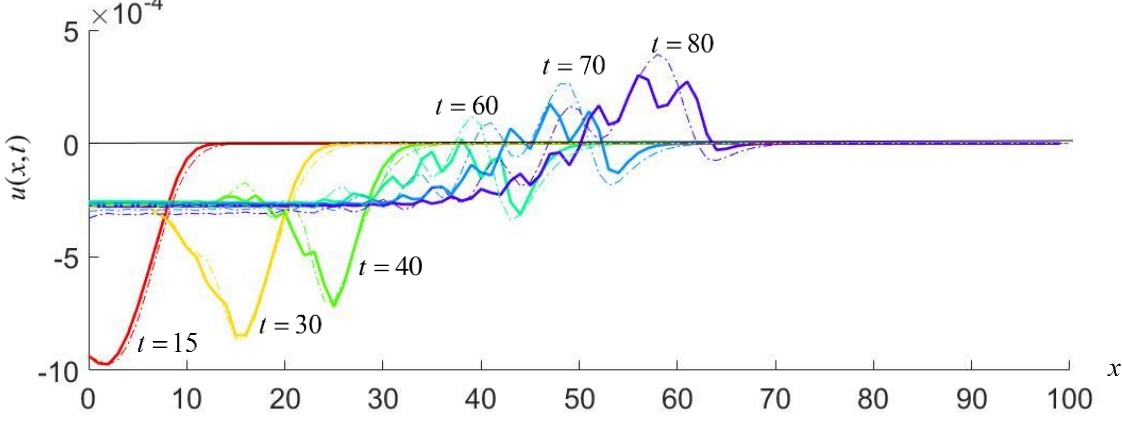

**Figure 8: Displacement $u(x,t)$ at different time moments versus the horizontal coordinate $x$ for the tension-compression harmonic impulse: numerical solutions for the discrete chain (solid line) and the bimodular rod (dash-dot line).**





**Conclusion**

This paper analysed the response of the discrete chain of bilinear oscillators and the bimodular rod subjected to several types of external harmonic excitation. The phenomenon of sign inversion of the displacement consisting of the gradual change of displacement sign for large times is observed for both the discrete chain and the bimodular rod under the tension-compression

impulse. It suggests that the collision between the two wave fronts corresponding to compression and tension phases has a considerable effect on the dynamic behaviour of the bilinear material.

It is anticipated that this observation may play an important role in geophysical and exploration applications, making it possible to detect bilinearity and thus obtain additional information on the composition and structure of the Earth's crust.

**Competing interests**

The authors declare that they have no conflict of interest.

**Acknowledgement**

The authors are grateful to Prof. Efim Pelinovsky and Dr. Andrey Radostin for discussing wave propagation in the bimodular media. The first author also gratefully acknowledges the scholarship support from The University of Western Australia (APA).

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
