# Peer review of "Analysis of Wave Propagation in a Discrete Chain of Bilinear Oscillators"

_Nonlinear Processes in Geophysics, 2016_

## Referee Comment (RC1) · Anonymous Referee #1 · 14 Feb 2017

**On the manuscript "Analysis of wave propagation in a discrete chain of bilinear oscillators" by M. Kuznetsova et al.**

The article presents the numerical results for a chain of masses connected by bilinear springs. The results are compared with those obtained in continuous approximation.

This is a potentially interesting topic. However, the manuscript is short and looks rather as an extended abstract. Too many details are omitted to enable a reliable evaluation. Still let me formulate several questions:

1. Even the basic boundary condition is not clear. The notation $H$ is not defined. If is the Heaviside step, $F(t)$ is non-zero for all $t < 2\pi / \omega$, including $t < 0$. Evidently, only t >0 is considered but it is never explicitly mentioned. Usually such impulses are represented as a difference of two Heaviside steps.
2. Since the wave changes its sign, then according to (1) the solution can not be analytical at Delta$U = 0$, and some matching conditions should be added at these points, at least for figure 3 where the front and tail of the impulse meet at some point.
3. For sec.6 where a continuous "rod" is considered. Since the numerical discretization (delta x =1) is the same as for the previous mass-spring model is used, what is the real difference between the two models? No surprise that in figs. 7 and 8 the results are identical.
4. Finally, what is the main physical result of this article? It is obvious that if the front propagates faster, the pulse is elongated (Fig. 2) and vice versa (fig.3). Can the authors add at least one physical system with specific estimates as an example?
5.  In its present version, the paper rather looks as a numerical exercise. If it is intended to be published as a full-fledged paper, I can not recommend it for NPG in this form.

---

## Referee Comment (RC2) · Anonymous Referee #2 · 17 Feb 2017

The authors have presented interesting results of a study on wave propagation in a discrete chain of bilinear oscillators for the case of rather large value of the parameter characterizing the difference between tension and compression moduli. The paper is worth to be published after taking into account the items that follow.

1. Please emphasize the differences with the paper [Gavrilov S.N. and Herman G.C., 2012].

2. What is the goal of this research?

3. It is necessary to note that in the case "tension – compression" the analytical solution exists for the stiffness ratio a«1, when the order of the Eq. (11) can be reduced and a solution with shock front exists, see [Naugolnykh, K., & Ostrovsky, L. (1998). Nonlinear wave processes in acoustics. Cambridge University Press]. In this connection, please

refine the peculiarity of the presented study.

4. Please estimate the product of the characteristic wave number and the length of the spring.

5. What methods of numerical simulations were used?

Typographical mistakes and other minor corrections

1. What is the correct notation for masses m or M?

2. The same for $\omega$ and $\Omega$.

3. Is the frequency of the applied force $\omega$ in the Table dimensionless?

4. One Heaviside function is used in the Eq. (8) instead of difference of two Heaviside functions.

Please also note the supplement to this comment:
http://www.nonlin-processes-geophys-discuss.net/npg-2016-80/npg-2016-80-RC2-supplement.pdf

---

## Author Response (AR1)

**Analysis of Wave Propagation in a Discrete Chain of Bilinear Oscillators**

Maria S. Kuznetsova[1], Elena Pasternak[2], Arcady V. Dyskin[1]

[1]School of Civil, Environmental and Mining Engineering, The University of Western Australia, Perth, 6009, Australia
[2]School of Mechanical and Chemical Engineering, The University of Western Australia, Perth, 6009, Australia

*Correspondence to*: Maria S. Kuznetsova (maria.kuznetsova@research.uwa.edu.au)

**Abstract.** The process of wave propagation in the discrete chain of bilinear oscillators subjected to several types of external harmonic excitation has been analysed. The phenomenon of sign inversion of the displacement is observed for tension-compression excitation. The solution for wave propagation in a continuous 1D bimodular rod is developed and the numerical results are compared.

**1 Introduction**

In this paper, we analyse the process of wave propagation in a chain of bilinear oscillators – discrete masses connected by springs having different stiffnesses in tension and compression. Due to their simplicity, discrete chains of bilinear oscillators have often been used in the problems related to non-linear vibrations of mechanical systems, such as vibrations in suspension bridges (De Freitas and Viana, 2004) and in the systems with the so-called fatigue cracks (Rivola and White, 1998; Ohara et al., 2007; Peng et al., 2008). Bilinear oscillators were also used in mathematical modelling of seismic isolation systems (Skinner et al. 1993; Chang et al., 2002). Layered rocks and rocks with a single set of open fractures obviously exhibit bilinear properties whereby the modulus in compression is higher than the modulus in tension due to the closure of interlayer gaps and fractures in compression.

The behaviour of the bilinear oscillators has been recently studied in (Dyskin et al., 2012; Dyskin et al., 2014; Guzek et al., 2016) for a limiting case of an infinite stiffness in compression. However, a general case of a discrete chain of bilinear oscillators has never been studied with respect to the mechanical wave propagation, which is why it has been decided to numerically investigate the response of the bilinear system that could represent a continuous bimodular medium. We focus on a conservative system; for the effects of damping in bilinear oscillators see (Shaw and Holmes, 1983; Natsiavas, 1990a, 1990b; Liu et al., 2015; Dyskin et al., 2012; Klepka et al., 2015, Guzek et al., 2016).

The purpose of the present work is to study the response of a discrete system of bilinear oscillators loaded by an external harmonic force, especially for the case of the large difference between spring stiffnesses in tension and compression.  In order to compare the chain of bilinear oscillators with its homogenised counterpart, we also considered a continuous 1D bimodular rod and developed a solution for its wave equation. In doing so, we will not restrict ourselves to small difference in stiffnesses, thus providing a more general analysis than the ones presented in (Naugonlykh and Ostovsky, 1998), (Gavrilov and Herman, 2012).

**2 Mathematical formulation**

We consider an infinite chain of masses and bilinear springs, where masses  $M$ are supposed to be identical, springs have the length $L_s$  and the stiffness described in the following formula

$$K(U) = \begin{cases} K_0(1-a) & \text{for } \Delta U \geq 0 \\ K_0(1+a) & \text{for } \Delta U < 0 \end{cases} \quad \text{},$$

(1)

Here, $U(X,T)$ is the displacement, $K_0$ is the average stiffness of the bilinear spring, $a$ is the stiffness ratio, and $\Delta U$ is the difference of displacement of two adjacent masses, that is the displacement of each spring. The mass-spring chain is fixed at the right end (Fig.1) and loaded by an external force $F(T)$ from the left end.

[Figure]

[Figure]

**Figure 1: Elastic mass-spring chain.**

By introducing the Lagrangian  $L = T - V$, which is the difference between the total kinetic and strain energy of the system where

$$E_k = \Sigma \frac{1}{2} M \dot{U}_i \text{ and } V = \Sigma \frac{1}{2} K_i (U_{i+1} - U_i)^2, \ i = 1..N,$$

(2)

We obtain the governing equation of the longitudinal motion of $i$-th mass

$$M\ddot{U}_i + (K_i + K_{i-1})U_i - K_{i-1}U_{i-1} - K_iU_{i+1} = F_i \quad \text{}$$

(3)

where $i = 1$ corresponds to the first mass from the left end. Since loading is applied to the left end, it follows that

$$F_i = \begin{cases} F(T) & \text{for } i = 1 \\ 0 & \text{otherwise} \end{cases} \quad \text{}.$$

(4)

Here and in what follows, consider harmonic external loading of the type $F(T) = F_0 \sin(\Omega T)$ where $F_0$ denotes any multiplier in front of the harmonic function and $\Omega$ denotes the external excitation frequency.

We rewrite the equation of motion (3) in terms of dimensionless displacement $u$ and dimensionless time $t$

$$u = \frac{\Omega_0}{c} U, \quad t = \Omega_0 T, \quad \text{}$$

(5)

where $\Omega$ $\Omega_0$ is the basic frequency of the bilinear oscillator $\Omega = \frac{2\pi}{T_0}$ $\Omega_0 = \sqrt{\frac{K_0}{M}}$ , $c$ is the sound velocity in the discrete

chain $c = L_s\sqrt{\frac{K}{M}}$ . This yields

$$\ddot{u}_i + (k_i + k_{i-1})u_i - k_{i-1}u_{i-1} - k_i u_{i+1} = F_i$$

$$\ddot{u}_i + (k_i + k_{i-1})u_i - k_{i-1}u_{i-1} - k_i u_{i+1} = f_i \qquad (6)$$

5    Here, $f_i$ $F_i$ and $k_i$ are the dimensionless forces and stiffnesses of the springs, respectively:

$$f_i = \frac{F_0}{M\Omega_0 c}\sin(\omega t) \text{ for } i=1, \; f_i = 0 \text{ otherwise}, \; k_i = \frac{K_i}{M\Omega_0^2}, \; \omega \text{ is the dimensionless excitation frequency } \omega = \frac{\Omega}{\Omega_0}. (7)$$

$$F_i = \frac{\tilde{F}_i}{M\Omega c}; k_i = \frac{K_i}{M\Omega^2} \qquad (7)$$

Without loss of generality, we adopt that the springs are stiffer in compression and obtain higher dimensionless compressive stiffness $k_c = 1 + a$ and lower tensile stiffness $k_t = 1 - a$ for $a > 0$., and by introducing the stiffness ratio $a$, obtain higher

10    compressive stiffness $k_c = 1 + a$ and lower tensile stiffness $k_t = 1 - a$.

The system is initially assumed to be at rest, i.e. $u_i(0) = \dot{u}_i(0) = 0$.

**3 Mechanical parameters of the discrete mass-spring chain**

All the numerical results presented in the paper are obtained for the following dimensionless parameters listed in the table below: dimensional and dimensionless parameters:

| | Notation | Value |
|---|---|---|
| Total number of masses | $N$ | 100 |
| Length of the spring | $l_s = \frac{\Omega_0}{c}L_s$ $l_s$ | 1 m |
| Stiffness ratio | $a$ | ⅓ |
| Amplitude of the applied force | $f_0 = \frac{F_0}{M\Omega_0 c}$ $F_0$ | $10^{-4}$ |
| Frequency of the applied force | $\omega$ | 0.25 |

15    **4 Impulse harmonic excitation**

In the analysis of wave propagation caused by initial excitation, simple harmonic or sinusoidal waves are of substantial interest. Due to its simplicity, let us analyse the case of a harmonic impulse first. The external loading subjected to the left end of the chain and is described as follows

$$f(t) = \pm f_0 H(t) H\left(\frac{2\pi}{\omega} - t\right)\sin(\omega t) \quad F(t) = \pm F_0 H\left(\frac{2\pi}{\omega} - t\right)\sin(\omega t)$$

20    $$(8)$$

where $H(t)$ is the Heaviside function. An explicit Runge-Kutta method with the time step $\Delta t = 10^{-3}$ is used for solving the system of $N$ bilinear ODEs (6) in Sect. 4.

**4.1 Compression-tension harmonic impulse**

[revised manuscript text omitted]

**6 Comparison with another numerical model and analytical solution**

In this section, we want to compare the numerical results for the discrete chain of bilinear oscillators with its homogenised counterpart, a continuous 1D bimodular rod, subjected to the same boundary conditions. In order to ensure whether the discrete chain with the given parameters can be considered as a continuum, let us estimate the dimensionless wave length $\lambda$

10  $\lambda = \dfrac{c_t \pi}{\omega} \approx 10.26$

The obtained wave length $\lambda$ is much greater than the spring length $l_s$, assumed to be equal to 1 (see Table 1), which is why the continuum approximation becomes possible. This will be done in order to check whether a numerical solution of the corresponding continuous problem can be accurate.

15   The wave equation for a 1D rod made of a bimodular material reads

$$\left(E_0 - e\,\mathrm{sgn}\left(\dfrac{\partial U}{\partial X}\right)\right)\dfrac{\partial^2 U}{\partial X^2} = \rho\dfrac{\partial^2 U}{\partial T^2} \quad \left(E - e\,\mathrm{sgn}\left(\dfrac{\partial U}{\partial X}\right)\right)\dfrac{\partial^2 U}{\partial X^2} = \rho\dfrac{\partial^2 U}{\partial T^2}$$

(11)

where $U, X, T$ are displacement, coordinate along the rod and time respectively, $\rho$ is the specific mass, $E_0$  is the "average" Young's modulus, and $e$ is the difference between Young's moduli in tension $E_t = E_0 - e$  and in
20  compression $E_c = E_0 + e$ .

In the dimensionless form, Eq. (11) reads

$$\left(1 - a\,\mathrm{sgn}\left(\dfrac{\partial u}{\partial x}\right)\right)\dfrac{\partial^2 u}{\partial x^2} = \dfrac{\partial^2 u}{\partial t^2}$$

(12)

where $x = \dfrac{\Omega}{\sqrt{E/\rho}}X$; $\quad t = \Omega T$; $\quad a = \dfrac{e}{E}$; $\quad u = \dfrac{\Omega}{\sqrt{E/\rho}}U$; $\quad \Omega = \dfrac{2\pi}{T_0}$ and $T_0$ is the duration of the applied external impulse.

The analytical solution for the compression-tension excitation of the frequency $\omega = 1$ described in Sect. 6.1 has been derived
25  in (Gavrilov and Herman, 2012) and was later extended for the arbitrary dimensionless excitation frequency in (Kuznetsova et al., 2016).

Numerical results are obtained by solving Eq. (12)  using the explicit central difference scheme.

To match the results obtained for the discrete chain of bilinear oscillators, spatial and time discretisation is chosen to be the same ($\Delta x = 1$ and $\Delta t = 10^{-3}$, respectively) and all other parameters being used from the Table 1.

**6.1 Compression-tension harmonic impulse**

The displacements $u(x,t)$ for various times are plotted in Fig. 7, which includes the analytical solution (bold dashed line) and numerical results for the discrete chain and the 1D bimodular rod (solid and dash-dot lines, respectively). One can observe that the three approaches show good agreement at the wave front and a slight discrepancy behind the wave front, which is typical for the second-order finite difference schemes (Kukudzhanov, 2013).

[Figure]

**Figure 7: Displacement $u(x,t)$ at different time moments versus the horizontal coordinate $x$ for the compression-tension harmonic impulse: analytical solution (dashed line), numerical solutions for the discrete chain (solid line) and the bimodular rod (thin dash-dot line).**

**6.2 Tension-compression harmonic impulse**

As the analytical solution does not exist for this case, only numerical results are presented. Figure 8 shows the numerical results for the displacements $u(x,t)$ for the discrete chain (solid line) and the 1D bimodular rod (dash-dot line). It is interesting to note that with all parameters being equal, the discrete chain generally exhibits lower displacements throughout the entire solution. This discrepancy may be explained by the insufficiently small spatial step for the rod since it is assumed it to be equal to the length of the springs in the discrete chain which equals 1.

[Figure]

**Figure 8: Displacement $u(x,t)$ at different time moments versus the horizontal coordinate $x$ for the tension-compression harmonic impulse: numerical solutions for the discrete chain (solid line) and the bimodular rod (dash-dot line).**

The direct comparison with numerical solution (with step over $x$ equal to 1) of the partial differential equation corresponding to the continuous rod demonstrates that the solutions are close. This indicates the possibility to solve the corresponding partial differential equation numerically despite the presence of discontinuous coefficients.

**Conclusion**

This paper analysed the response of the discrete chain of bilinear oscillators and the bimodular rod subjected to several types of external harmonic excitation. To the best of our knowledge, wave propagation in bilinear oscillators with large stiffness ratio has never been considered before. The phenomenon of sign inversion of the displacement consisting of the gradual change of displacement sign for extended times is observed for both the discrete chain and the bimodular rod under the tension-compression impulse. It suggests that the collision between the two wave fronts corresponding to compression and tension phases has a considerable effect on the dynamic behaviour of the bilinear material.

It is anticipated that this observation may play an important role in geophysical and exploration applications, making it possible to detect bilinearity and thus obtain additional information on the composition and structure of the Earth's crust.

**Competing interests**

The authors declare that they have no conflict of interest.

**Acknowledgement**

The authors are grateful to Prof. Efim Pelinovsky and Dr. Andrey Radostin for discussing wave propagation in the bimodular media. The first author also gratefully acknowledges the support provided by the Australian Government and the University of Western Australia in the form of the International Postgraduate Research Scholarship.scholarship support from The University of Western Australia (APA).

The authors are grateful for the important suggestions provided by the reviewer. All the suggestions have been taken into account and the manuscript has been changed accordingly.

**Suggestions and alterations**

| | |
|---|---|
| **1** | Even the basic boundary condition is not clear. The notation $H$ is not defined. If is the Heaviside step, $F$(t) is non-zero for all $t < 2\pi / \omega$, including $t < 0$. Evidently, only t >0 is considered but it is never explicitly mentioned. Usually such impulses are represented as a difference of two Heaviside steps. |
| | Corrected in Sect. 4 and Sect.5:

$f(t) = \pm f_0 H(t) H\left(\dfrac{2\pi}{\omega} - t\right) \sin(\omega t)$

$f(t) = \pm f_0 H(t) \sin(\omega t)$ |
| **2** | Since the wave changes its sign, then according to (1) the solution can not be analytical at Delta$U = 0$, and some matching conditions should be added at these points, at least for figure 3 where the front and tail of the impulse meet at some point. |
| | As pointed out in the paper we only consider a chain of bilinear oscillators. Therefore it is just a system of ordinary differential equations with discontinuous coefficient at the function. This can be solved by usual numerical schemes as shown in numerous literature cited in Introduction. Furthermore, direct comparison with numerical solution (with step over x equal to 1) of the corresponding partial differential equation corresponding to the continuous rod demonstrate that the solutions are close. This indicates the possibility to solve the corresponding partial differential equation numerically dispite the presence of discontinuous coefficients. |
| **3** | For sec.6 where a continuous "rod" is considered. Since the numerical discretization (delta x =1) is the same as for the previous mass-spring model is used, what is the real difference between the two models? No surprise that in figs. 7 and 8 the results are identical. |
| | We added the following sentence at the beginning of Sect. 6:
"In this section, we want to compare the numerical results for the discrete chain of bilinear oscillators with its homogenised counterpart, a continuous 1D bimodular rod, subjected to the same boundary conditions. This will be done in order to check whether a numerical solution of the corresponding continuous problem can be accurate."

We then added the following sentence at the end of subsection 6.2.
"The direct comparison with numerical solution (with step over $x$ equal to 1) of the partial differential equation corresponding to the continuous rod demonstrates that the solutions are close. This indicates the possibility to solve the corresponding partial differential equation numerically despite the presence of discontinuous coefficients." |
| **4** | Finally, what is the main physical result of this article? It is obvious that if the front propagates faster, the pulse is elongated (Fig. 2) and vice versa (fig.3). Can the authors add at least one physical system with specific estimates as an example? |

We added the following in the next-to-last paragraph of the introduction:
"The purpose of the present work is to study the response of a discrete system of bilinear oscillators loaded by an external harmonic force. Attention has been given to a case of the large difference between spring stiffnesses in tension and compression."

We also added the following sentence at the end of the first paragraph of the introduction. "Layered rocks and rocks with a single set of open fractures obviously exhibit bilinear properties whereby the modulus in compression is higher than the modulus in tension due to the closure of interlayer gaps and fractures in compression."

**Answers on the review for NPG-2016-80 by M. Kuznetsova, E. Pasternak and A. Dyskin, "Analysis of Wave Propagation in a Discrete Chain of Bilinear Oscillators"**

The authors are grateful for the perusal and important suggestions provided by the reviewer. All the suggestions have been taken into account and the manuscript has been changed accordingly.

**Changes**

1. Please emphasize the differences with the paper [Gavrilov S.N. and Herman G.C., 2012].

    *Response*
    We added the following sentences in the last paragraph of the introduction:
    "In order to compare the chain of bilinear oscillators with its homogenised counterpart, we also considered a continuous 1D bimodular rod and developed a solution for its wave equation. In doing so, we will not restrict ourselves to small difference in stiffnesses, thus providing a more general analysis than the ones presented in (Naugonlykh and Ostovsky, 1998), (Gavrilov and Herman, 2012)."

2. What is the goal of this research?
    *Response*
    We added the following sentence in the next-to-last paragraph of the introduction:
    "The purpose of the present work is to study the response of a discrete system of bilinear oscillators loaded by an external harmonic force, especially for the case of the large difference between spring stiffnesses in tension and compression."

3. It is necessary to note that in the case "tension – compression" the analytical solution exists for the stiffness ratio $a<<1$, when the order of the Eq. (11) can be reduced and a solution with shock front exists, see [Naugolnykh, K., & Ostrovsky, L. (1998). *Nonlinear wave processes in acoustics*. Cambridge University Press]. In this connection, please refine the peculiarity of the presented study.
    *Response*
    It should be noted that approximate analytical and numerical results for 1D bimodular rod are presented in (Naugonlykh and Ostovsky, 1998). However, they were obtained for a considerable limitation on the stiffness ratio being close to $\ll 1$, whereas we purposefully consider a case of the large difference between moduli in tension and compression as the most representative example of collision between tensile and compressive wave fronts and conservation of energy and the same time.
    We added the following sentence in the last paragraph of the introduction:
    "In doing so, we will not restrict ourselves to small difference in stiffnesses, thus providing a more general analysis than the ones presented in (Naugonlykh and Ostovsky, 1998), (Gavrilov and Herman, 2012)."

4. Please estimate the product of the characteristic wave number and the length of the spring.
    *Response*
    We added the following information in the beginning of Sect. 6:
    In order to ensure whether the discrete chain with the given parameters can be considered as a continuum, let us estimate the dimensionless wave length $\lambda$

$$\lambda = \frac{c_t \pi}{\omega} \approx 10.26$$

The obtained wave length $\lambda$ is much greater than the spring length $l_s$, assumed to be equal to 1 (see Table 1), which is why the continuum approximation becomes possible.

5. What methods of numerical simulations were used?
*Response*
We added the following sentence in Sect. 4:
"An explicit Runge-Kutta method with the time step $\Delta t = 10^{-3}$ is used for solving the system of $N$ bilinear ODEs (6) in Sect. 4."
We also have the following information in Sect. 6:
"Numerical results are obtained by solving Eq. (12) using the explicit central difference scheme. To match the results obtained for the discrete chain of bilinear oscillators, spatial and time discretisation is chosen to be the same ($\Delta x = 1$ and $\Delta t = 10^{-3}$, respectively) and all other parameters being used from the Table 1."

**Typographical mistakes**

1. What is the correct notation for masses $m$ or $M$?
*Response*
Corrected in Sect. 2:
"We consider an infinite chain of masses and bilinear springs, where masses $M$ are supposed to be identical, springs have the length $L_s$ and the stiffness described in the following formula"

2. The same for ω and Ω.
*Response*
Corrected in Sect. 2:
"Here and in what follows, consider harmonic external loading of the type $F(T) = F_0 \sin(\Omega T)$ where $F_0$ denotes any multiplier in front of the harmonic function and $\Omega$ denotes the external excitation frequency."

"$\omega$ is the dimensionless excitation frequency $\omega = \frac{\Omega}{\Omega_0}$."

3. Is the frequency of the applied force ω in the Table dimensionless?
*Response*
Yes. We changed its definition in Sect. 2 and altered the Table in Sect. 3:
"$\omega$ is the dimensionless excitation frequency $\omega = \frac{\Omega}{\Omega_0}$."

"All the numerical results presented in the paper are obtained for the following dimensionless parameters:"

4. One Heaviside function is used in the Eq. (8) instead of difference of two Heaviside functions.
*Response*
Corrected in Sect. 4 and Sect.5:

$$f(t) = \pm f_0 H(t) H\left(\frac{2\pi}{\omega} - t\right) \sin(\omega t), \quad f(t) = \pm f_0 H(t) \sin(\omega t).$$